# Determining factors of neonatal mortality in Ethiopia: An investigation from the 2019 Ethiopia Mini Demographic and Health Survey

**Abay Sahile**[ORCID]*, **Dereje Bekele**[◕], **Habtamu Ayele**[◕]

Department of Statistics, Madda Walabu University, Robe, Oromia, Ethiopia

◕ These authors contributed equally to this work.
* abysahile@gmail.com

## Abstract

### Background

Neonatal mortality is the probability of dying during the first 28 days of life. Of approximately 5 million children who die in the first year of life in the world, about 3 million are within their first 28 days of birth. In Ethiopia, the neonatal mortality rate is high about 37 per 1000 live births, and the factors are not well documented. Then, this study aimed to determine the key factors that have a significant influence on neonatal mortality.

### Methods

A total of 5753 neonatal mortality-related data were obtained from Ethiopia Mini Demographic and Health Survey (2019) data. A frequency distribution to summarize the overall data and Binary Logistic Regression to identify the subset of significant risk factors for neonatal mortality were applied to analyze the data.

### Results

An estimated 36 per 1000 live children had died before the first 28 days, with the highest in the Benishangul Gumuz region (15.9%) and the lowest in Addis Ababa (2.4%). From the Binary logistic regression analysis, the odds ratio and 95% CI of age 25–34 (OR = 0.263, 95% CI: 0.106–0.653), Afar (OR = 0.384, 95% CI: 0.167–0.884), SNNPR (OR = 0.265, 95% CI: 0.098–0.720), Addis Ababa (OR = 5.741, 95% CI: 1.115–29.566), Urban (OR = 0.253, 95% CI: 0.090, 0.709), toilet facility (OR = 0.603, 95% CI: 0.404–0.900), single birth (OR = 0.261, 95% CI: 0.138–0.495), poorest (OR = 10.573, 95% CI: 2.166–51.615), poorer (OR = 19.573, 95% CI: 4.171–91.848), never breastfed (OR = 35.939, 95% CI: 25.193–51.268), public health delivery (OR = 0.302, 95% CI: 0.106–0.859), private health facility (OR = 0.269, 95% CI: 0.095–0.760).

### Conclusion

All regional states of Ethiopia, specially Benishangul Gumuz, and the Somali region must take remedial actions on public health policy, design strategies to improve facilities, and

**Data Availability Statement:** The data set used for this study cannot be shared or distributed publicly either directly or within any tool or dashboard. This was restricted by the data privacy regulations of

DHS. The data must not be passed on to others without the written consent of DHS. To connect with the data archivist contact archive@dhsprogram.com.

**Funding:** This study was financially supported by Madda Walabu University, Ethiopia. However, the funders had no role in study design, data collection and analysis, decision to publish, or preparation of the manuscript.

**Competing interests:** The authors declare that we have no competing interests.

improve the capacities of stakeholders living in their region toward those major factors affecting neonatal mortality in the country.

## Background

Child death during the first month (28 days) of life is called neonatal mortality. For the child, the period is the most vulnerable time to survive. Globally, in 2020 an estimated 5 million under-five age children died, and approximately half (2.4 million) of these occurs within the first 28 days of life [1]. Based on the place where the child is born, there are variations in neonatal mortality. It becomes an important public health issue in developing countries like Sub-Saharan Africa and Central and Southern Asia. The rate was high, about 27 deaths per 1000 live births in Sub-Saharan Africa, covering 43% of global newborn deaths, which shows that the neonates in the region were 10 times more likely to die than the neonates in developed countries [2].

From 1990, different important progress was done to survive the children in the world. Due to this, the rate of deaths of neonatal mortality declined by 54%, from 37 in 1990 to 17 per 1000 live births in 2020. However, this declining rate was slower as compared to the under-five mortality rate, which declined by 61% [3]. This is maybe due to most previous studies conducted focused on the under-five mortality rate rather than the neonatal mortality rate [4].

Ethiopia has the third-highest reported number of newborn deaths in Africa, and the fifth number of deaths in the world [5]. Approximately 42% of the under-5 mortality in Ethiopia is attributable to neonatal deaths [6]. From the report of Ethiopia Mini Demographic and Health Surveys (EMDHS), the trends of neonatal mortality declined from 39 in 2005 to 29 deaths per 1000 lived in 2016. However, the rate slightly increased to 33 deaths per 1000 live births in 2019 [7]. Different risk factors associated with neonatal mortality like child breastfed status, family place of residence, place of delivery, families wealth index, toilet facility, sepsis, asphyxia, birth injury, tetanus, preterm birth, and congenital malformations were adopted from the literature [1, 8–10]. However, the highly significant risk factors of neonatal mortality were not well documented in Ethiopia. Therefore, this study was conducted to list the significant risk factors that are highly associated and slightly increase the number of neonatal mortality in Ethiopia.

## Material and methods

### Data source

The data set used for this study was obtained from the Ethiopia Mini Demographic and Health Survey (2019). It was implemented by the Ethiopian Public Health Institute (EPHI), and for this study, we were authorized to use it from the Demographic and Health Surveys (DHS) Program.

### Sampling techniques

The 2019 EMDHS sample was stratified and selected in two stages. Each region was stratified into urban and rural areas, yielding 21 sampling strata. Samples of enumeration areas (EAs) were selected independently in each stratum in two stages. In the first stage, a total of 305 EAs (93 in urban areas & 212 in rural areas) were selected with probability proportional to EA size (based on the 2019 Population and Housing Census frame) & with independent selection in

each sampling stratum. A household listing operation was carried out in all selected EAs from January through April 2019. The resulting lists of households served as a sampling frame for the selection of households in the second stage. In the second stage of selection, a fixed number of 30 households per cluster were selected with an equal probability of systematic selection from the newly created household listing. All women aged 15–49, who were either permanent residents of the selected households or visitors who slept in the household the night before the survey, were eligible to be interviewed [7].

## Study variables

The response variable of interest for this study is the neonatal mortality status of children, before reaching the first 28 days of life. It was dichotomized into No for not died children, and Yes for those who died within 28 days after birth. The independent variables like mother's age at birth, region, place of residence, mother's education level, source of drinking water, type of toilet facility, religion, sex of household head, wealth index, total children ever born, current marital status, birth order number, type of birth, sex of a child, breastfeeding status and place of delivery were adopted from literature.

## Statistical models

The data for this study was extracted from EMDHS 2019, and analyzed using SPSS version 25 statistical software. Descriptive statistics like the measure of frequency (count and percent), mean and standard deviation to summarize the main features of the data, and inferential statistics multiple logistic regression were applied. The binary logistic regression is the type of regression that was applied to identify the potential subset of significant risk factors for neonatal mortality in Ethiopia. In logistic regression, the response variable is a logit, which is the natural logarithm of the odds. The model is given as:

$$\mathbf{Log(Odds)} = \mathbf{logit(Pi)} = \mathbf{ln}\left[\frac{\mathbf{Pi}}{\mathbf{1 - Pi}}\right] = \mathbf{X^T\beta} \tag{1}$$

Where X = independent variables and β = regression coefficient and $Pi$ = probability of child death before the first 28 days of life.

## Results

Of the total of 5753 births, about 208 (3.6%) have died before their first 28 days of life. The highest number about 33 (15.9%) and the lowest about 5 (2.4%) of these deaths were observed in Benishangul Gumuz and Addis Ababa respectively. The majority, about 91 (43.8%), 199 (95.7%), 120 (57.7%), 92 (44.2%), and 128 (61.5%) neonatal mortality were those whose mothers aged 25–34, who were from rural communities, uneducated family, have a piper drinking water resource and those whose family have no toilet facility respectively. For the poorest wealth index family, married, a single type of birth, never breastfed, and born at home; the largest number of neonatal deaths were about 84 (40.4%), 189 (90.9%), 181 (87.0%), 127 (61.1%), 133 (63.9%) respectively were recorded. Whereas, for those whose families were from urban, with higher education level, rainwater users, the richest, single in marital status and delivered their baby at a private health facility; too small number about 9 (4.3%), 5 (2.4%), 2 (1.0%), 4 (1.9%), 1 (0.5%) and 8 (3.8%) neonatal mortality were observed respectively. In general, the average number of total children ever born and the number of birth orders are approximately similar; about 4 in neonatal deaths with 2.836 and 2.806 respectively (Table 1).

**Table 1. Descriptive summary of neonatal death-related characteristics, EMDHS (2019).**

| Variables | Categories | Neonatal death | Total birth |
|---|---|---|---|
| | | n (%) | n (%) |
| | | n = 208 (3.6%) | n = 5753 |
| Mothers age at birth | 15–24 | 69 (33.2%) | 1439 (25.0%) |
| | 25–34 | 91 (43.8%) | 3097 (53.8%) |
| | 35–44 | 35 (16.8%) | 1104 (19.2%) |
| | 44+ | 13 (6.3%) | 113 (2.0%) |
| Region | Tigray | 9 (4.3%) | 454 (7.9%) |
| | Afar | 19 (9.1%) | 652 (11.3%) |
| | Amhara | 19 (9.1%) | 511 (8.9%) |
| | Oromia | 27 (13.0%) | 719 (12.5%) |
| | Somali | 30 (14.4%) | 637 (11.1%) |
| | Benishangul | 33 (15.9%) | 530 (9.2%) |
| | SNNPR | 15 (7.2%) | 660 (11.5%) |
| | Gambela | 16 (7.7%) | 450 (7.8%) |
| | Harari | 20 (9.6%) | 447 (7.8%) |
| | Addis Adaba | 5 (2.4%) | 291 (5.1%) |
| | Dire Dawa | 15 (7.2%) | 402 (7.0%) |
| Place of residence | Urban | 9 (4.3%) | 1291 (22.4%) |
| | Rural | 199 (95.7%) | 4462 (77.6%) |
| Highest educational level | No education | 120 (57.7%) | 3154 (54.8%) |
| | Primary | 76 (36.5%) | 1822 (31.7%) |
| | Secondary | 7 (3.4%) | 476 (8.3%) |
| | Higher | 5 (2.4%) | 301 (5.2%) |
| Source of drinking water | Piped water | 92 (44.2%) | 2577 (44.8%) |
| | Tube well water | 38 (18.3%) | 869 (15.1%) |
| | Surface from spring | 71 (34.1%) | 2021 (35.1%) |
| | Rainwater | 2 (1.0%) | 159 (2.8%) |
| | Bottled water | 5 (2.4%) | 127 (2.2%) |
| Type of toilet facility | Facility | 80 (38.5%) | 3212 (55.8%) |
| | No Facility | 128 (61.5%) | 2541 (44.2%) |
| Religion | Orthodox | 50 (24.0%) | 1612 (28.0%) |
| | Catholic | 1 (0.5%) | 32 (0.6%) |
| | Protestant | 23 (11.1%) | 1053 (18.3%) |
| | Muslim | 131 (63.0%) | 2974 (51.7%) |
| | Others | 3 (1.4%) | 82 (1.4%) |
| Sex of household head | Male | 166 (79.8%) | 4598 (79.9%) |
| | Female | 42 (20.2%) | 1155 (20.1%) |
| Wealth index | Poorest | 84 (40.4%) | 1985 (34.5%) |
| | Poorer | 59 (28.4%) | 1014 (17.6%) |
| | Middle | 34 (16.3%) | 805 (14.0%) |
| | Richer | 27 (13.0%) | 738 (12.8%) |
| | Richest | 4 (1.9%) | 1211 (21.0%) |
| Current marital status | Single | 1 (0.5%) | 31 (0.5%) |
| | Married | 189 (90.9%) | 5396 (93.8%) |
| | Widowed | 2 (1.0%) | 61 (1.1%) |
| | Divorced | 16 (7.7%) | 265 (4.6%) |

*(Continued)*

**Table 1.** (Continued)

| Variables | Categories | Neonatal death | Total birth |
|---|---|---|---|
| | | n (%) | n (%) |
| | | n = 208 (3.6%) | n = 5753 |
| Types of birth | Single | 181 (87.0%) | 5586 (97.1%) |
| | Multiple | 27 (13.0%) | 167 (2.9%) |
| Sex of child | Male | 135 (64.9%) | 2985 (51.9%) |
| | Female | 73 (35.1%) | 2768 (48.1%) |
| Duration of breastfeeding | Never breastfed | 127 (61.1%) | 378 (6.6%) |
| | Breastfed | 81 (38.9%) | 5375 (93.4%) |
| Place of delivery | Home | 133 (63.9%) | 2975 (51.7%) |
| | Public health facility | 67 (32.2%) | 2536 (44.1%) |
| | Private health facility | 8 (3.8%) | 242 (4.2%) |
| Total children ever born | (Mean, SD) | 4.26, 2.836 | 4.04, 2.480 |
| Birth order number | (Mean, SD) | 3.70, 2.809 | 3.68, 2.431 |

## Binary logistic regression analysis

From the Binary logistic regression result Table 2, at a 5% level of significance, mother's age at birth, region, place of residence, type of toilet facility, wealth index, total children ever born, birth order, type of birth, sex of the child, breastfeeding status and place of delivery were significantly associated to the neonatal mortality in Ethiopia.

Considering other risk factors constant, the odds of neonatal mortality who born from mothers aged between 25–35 (OR = 0.263; 95% CI: 0.106–0.653) and 35–44 (OR = 0.307, 95% CI: 0.127–0.743) is about 0.263 and 0.307 times less likely than the newborn from mothers aged above 44. For neonates born from urban lived mothers (OR = 0.253, 95% CI: 0.090–0.709), the odds of dying were about 0.253 less likely than those who were born from rural mothers. Compared to families of neonates who have a toilet facility (OR = 0.603, 95% CI: 0.404–0.900), the odds of neonates dying for those who do not have a toilet facility was about 0.603 less. Holding others constant, the odds of neonates dying in the poorest (OR = 10.573, 95% CI: 2.166–51.615), poorer (OR = 19.573, 95% CI: 4.171–91.848), middle (OR = 12.773, 95% CI: 2.638–61.859), and richer (OR = 12.562, 95% CI: 2.675–58.983) family were respectively about 10.573, 19.573, 12.773, and 12.562 times more likely than in those who have the richest family. Allowing others constant, an increase in one number of children ever born increases the odds of neonatal mortality by 1.507 times (OR = 1.507, 95% CI: 1.161–1.956). As the number of birth order increase by one, the odds of neonatal mortality decrease by 0.637 (OR = 0.637, 95% CI: 0.488–0.832). Compared to being multiple in birth type, the odds of dying for single in birth type neonates (OR = 0.261, 95% CI: 0.138–0.495) were about 0.261 less. The odds of dying male neonates (OR = 1.531, 95% CI: 1.083–2.166) were about 1.531 more likely than that of female neonates. Considering other constants, the odds of dying never-breastfeeding neonates (OR = 35.939, CI: 25.193–51.268) were about 36 times more likely than that of breastfed neonates. Regarding the place of delivery, the odds of dying neonates who were born at public health (OR = 0.302, CI: 0.106–0.859) and private health facilities (OR = 0.269, CI: 0.095–0.760) were 0.302 and 0.269 respectively times less likely than those who were born at home Table 2.

## Discussion

The general goal of this study was to identify risk factors associated with neonatal mortality in Ethiopia using a nationally representative sample. This study revealed numerous factors that

**Table 2. Multiple logistic regression results for neonatal mortality risk factors, EMDHS (2019).**

| Variables | AOR | 95% C.I for AOR | | P-Value |
|---|---|---|---|---|
| | | Lower | Upper | |
| Mothers age at birth (Ref: 44+) | | | | 0.004 |
| 15–24 | 0.411 | 0.146 | 1.156 | 0.092 |
| 25–34 | 0.263 | 0.106 | 0.653 | 0.004 |
| 35–44 | 0.307 | 0.127 | 0.743 | 0.009 |
| Region (Ref: Dire Dawa) | | | | 0.002 |
| Tigray | 0.436 | 0.136 | 1.398 | 0.163 |
| Afar | 0.384 | 0.167 | 0.884 | 0.025 |
| Amhara | 0.871 | 0.317 | 2.396 | 0.790 |
| Oromia | 0.711 | 0.314 | 1.609 | 0.413 |
| Somali | 0.617 | 0.281 | 1.355 | 0.229 |
| Benishangul | 1.165 | 0.517 | 2.624 | 0.712 |
| SNNPR | 0.265 | 0.098 | 0.720 | 0.009 |
| Gambela | 0.601 | 0.218 | 1.658 | 0.325 |
| Harari | 0.698 | 0.290 | 1.675 | 0.421 |
| Addis Adaba | 5.741 | 1.115 | 29.566 | 0.037 |
| Place of residence (Ref: Rural) | | | | |
| Urban | 0.253 | 0.090 | 0.709 | 0.009 |
| Highest educational level (Ref: Degree) | | | | 0.070 |
| No education | 0.472 | 0.139 | 1.598 | 0.228 |
| Primary | 0.745 | 0.226 | 2.454 | 0.628 |
| Secondary | 0.320 | 0.072 | 1.430 | 0.136 |
| Type of toilet facility (Ref: No facility) | | | | |
| Facility | 0.603 | 0.404 | 0.900 | 0.013 |
| Religion (Ref: Others) | | | | 0.240 |
| Orthodox | 2.516 | 0.648 | 9.767 | 0.183 |
| Catholic | 2.205 | 0.166 | 29.272 | 0.549 |
| Protestant | 1.376 | 0.362 | 5.227 | 0.639 |
| Muslim | 2.718 | 0.720 | 10.258 | 0.140 |
| Wealth index (Ref: Richest) | | | | 0.001 |
| Poorest | 10.573 | 2.166 | 51.615 | 0.004 |
| Poorer | 19.573 | 4.171 | 91.848 | 0.000 |
| Middle | 12.773 | 2.638 | 61.859 | 0.002 |
| Richer | 12.562 | 2.675 | 58.983 | 0.001 |
| Total children ever born | 1.507 | 1.161 | 1.956 | 0.002 |
| Birth order number | 0.637 | 0.488 | 0.832 | 0.001 |
| Types of birth (Ref: Multiple births) | | | | |
| Single birth | 0.261 | 0.138 | 0.495 | 0.000 |
| Sex of child (Ref: Female) | | | | |
| Male | 1.531 | 1.083 | 2.166 | 0.016 |
| Breastfeeding status (Ref: Breastfed) | | | | |
| Never breastfed | 35.939 | 25.193 | 51.268 | 0.000 |
| Place of delivery (Ref: Home) | | | | 0.046 |
| Public health facility | 0.302 | 0.106 | 0.859 | 0.025 |
| Private health facility | 0.269 | 0.095 | 0.760 | 0.013 |
| Constant | 0.062 | | | 0.030 |

were significantly associated with neonatal mortality. The residence place of mothers significantly determined the survival of neonates. Thus, the risk of neonatal death was less likely among neonates born from urban households compared to neonates born from rural households. This result was in line with the earlier studies conducted in Bangladesh [11], Nigeria [12], and China [13]. This is probably because families who live in urban areas are typically well-found with healthier infrastructure for health services that contain higher coverage with immunization and safe delivery as opposed to families who live in rural areas [14]. Neonatal mortality was less likely to happen among mothers who aged in the age group 25–44 years old compared to mothers who aged above 44 years old. This finding is analogous to the earlier study by Ribeiro [15], which reported that mothers aged younger than 15years or aged 35 above have experienced a higher risk of child mortality 21. The difference may be due to a higher risk of delivering high or low birth weight babies among older mothers and also large maternal age is linked with maternal health care and delivery complication [16]. Concerning toilet facilities, the result of this study indicated that the risk of experiencing neonatal death among neonates having families with toilet facilities was less likely compared to the neonates from families without toilet facilities. This finding is supported by the preceding study conducted in Pakistan [17]. Similarly, the risk of neonatal death was more likely to occur among neonates from the poorest family as compared to the richest family. This result is consistent with previous studies conducted in Nigeria [18], Kenya [19], and Sudan [20]. This occurs may be due to families with high-income levels might get necessary health services during pregnancy, delivery and after delivery, and the neonates might be provided better care than a family with low income [21]. Male neonates were more likely to die than female neonates. This is supported by the earlier studies done in Ethiopia by Mitiku, 2021 [22], and Nigeria by Ezeh et al, 2014 [23]. This is probably attributed to the protein and gene expressions in placentae of male and female fetuses are differ, especially during adverse conditions. In addition, the miRNAs play a significant role in the development of the fetus and the placenta by regulating their target genes, the miRNA could show downregulation in male and up-regulation in females fetuses. This makes females have a regular survival benefit than males [24]. Regarding the type of birth, the result of this study revealed that multiple births had the highest odds of neonatal death compared to single birth. This finding was agreed with the previous study done by Kibria et al in Afghanistan [25], and Kamal SMM in Bangladesh [26]. This may be due to the situation that twin gestation is commonly linked with prematurity, which was the leading cause of death in twins, next to twin-to-twin transfusion syndrome (TTTS) which leads to mortality [27]. Additionally, twin gestation frequently occurs with low birth weight which increases neonates' vulnerability to infection and decreases their immunity [28].

Breastfeeding status was strongly associated with a higher risk of neonatal death. This study showed that the odds of neonatal death were more likely to happen among neonates those never breastfed as compared to neonates who were breastfed. This result is aligned with the previous studies conducted in Ethiopia [22] and Ghana [29]. This may be attributed to the fact that breast milk delivers the perfect nutrition for neonates that helps to fight numerous communicable diseases, reinforce vital antibody systems, and also helps brain development [30]. Based on the total number of children ever born, the outcomes of this study showed that the odds of neonatal mortality increased as the total number of children ever born from families increased. This finding is matching with the previous study done by Adhikari and Sawangdee in Nepal in 2011 [31]. This could be a family with a small number of children provided better health services, and a balanced diet than a family with a large number of children [14]. Depending on the birth order number, the result of this study showed that the odds of experiencing neonatal mortality in newborn babies were reduced by 37% as the birth order

number of mothers increased. This finding was in line with the study conducted in Nigeria [32], which reported that a wider birth order number existing lower the mortality rate.

Finally, the findings of this study revealed that the risk of neonatal mortality was significantly associated with the place of delivery. Hence, neonates who delivered at a health facility had lower neonatal death compared with neonates who delivered at home. This finding is alike to a matched case-control study in Indonesia in 2016 [33], and a study conducted in India in 2022 [34], both authors reported that the risk of neonatal death was less likely for neonates who were delivered in health institutions than delivered at home. This may be occurred due to a lack of adequate maternal and newborn care at home. Consequently, the newborn baby might not take the immediate nutrition after birth and it is also difficult to fix complications immediately if there is a problem while the mother is delivered at home [22].

## Conclusions

In Ethiopia, the neonatal mortality rate was about 36 deaths per 1000 live, and it was highly observed in the Benishangul Gumuz region. Of various risk factors considered in this study, borning from a rural mother, the poorest family, and being non-breastfeeding of the child at birth highly caused the baby to die within 28 days.

## Acknowledgments

We are indebted to the Demographic and Health Survey Program for permitting us to use the data for this study.

## Author Contributions

**Conceptualization:** Abay Sahile.

**Data curation:** Abay Sahile, Dereje Bekele, Habtamu Ayele.

**Formal analysis:** Abay Sahile, Dereje Bekele, Habtamu Ayele.

**Methodology:** Abay Sahile, Dereje Bekele, Habtamu Ayele.

**Supervision:** Dereje Bekele.

**Writing – original draft:** Abay Sahile, Dereje Bekele, Habtamu Ayele.

**Writing – review & editing:** Abay Sahile, Dereje Bekele, Habtamu Ayele.

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
