## [Decision Letter · Decision Letter 0]

26 Oct 2022

PONE-D-22-11460Determining factors of Neonatal Mortality in Ethiopia: An investigation from the 2019 Ethiopia Mini Demographic and Health SurveyPLOS ONE

Dear Abay Sahile,

Thank you for submitting your manuscript to PLOS ONE. After careful consideration, we feel that it has merit but does not fully meet PLOS ONE’s publication criteria as it currently stands. Therefore, we invite you to submit a revised version of the manuscript that addresses the points raised during the review process.

Your article is good, and findings are instrumental for application and improvement in local context. 

We look forward to receiving your revised manuscript.

Kind regards,

Sidrah Nausheen, FCPS

Academic Editor

PLOS ONE

“This study was financially supported by Madda Walabu University, Ethiopia.”

“We would like to acknowledge Madda Walabu University of Research, Community Engagement, and Technology Transfer Vice President Office for the financial support. We are also indebted to the Demographic and Health Survey Program for permitting us to use the data for this study.”

“This study was financially supported by Madda Walabu University, Ethiopia.”

6. Thank you for stating the following in your Competing Interests section: 

“The authors declare that we have no competing interests.”

7. In your Data Availability statement, you have not specified where the minimal data set underlying the results described in your manuscript can be found. PLOS defines a study's minimal data set as the underlying data used to reach the conclusions drawn in the manuscript and any additional data required to replicate the reported study findings in their entirety. All PLOS journals require that the minimal data set be made fully available. For more information about our data policy, please see http://journals.plos.org/plosone/s/data-availability.

Reviewers' comments:

Reviewer's Responses to Questions

**Comments to the Author**

1. Is the manuscript technically sound, and do the data support the conclusions?

Reviewer #1: Yes

Reviewer #2: Yes

2. Has the statistical analysis been performed appropriately and rigorously? 

Reviewer #1: Yes

Reviewer #2: Yes

3. Have the authors made all data underlying the findings in their manuscript fully available?

Reviewer #1: Yes

Reviewer #2: Yes

4. Is the manuscript presented in an intelligible fashion and written in standard English?

Reviewer #1: Yes

Reviewer #2: Yes

5. Review Comments to the Author

Reviewer #1: Comment 1 : Page 5,Line 85: Heading Data Source, the information related to the study design of EMDHS is not written.

Comment 2 : Page 6,Line 114: The information related to the software used for analysis is not available.

Comment 3 : Page 10, Line 142 (Table 3): Table has too much information, it can be only restricted to Exp(B)/Odd Ratio values and 95 %Confidence Interval.

Comment 4 : Page 10 & 14 , Line 138 & 165 : remove empty round brackets.

Reviewer #2: This is a well written report, and the findings are useful. The findings are important to convey in local context for policy makers. I don't have specific comments, anyhow please carefully check the spelling and grammar.

.

---

## [Author Response · Author response to Decision Letter 0]

27 Nov 2022

Thank you for allowing us to submit the revised manuscript [PONE-D-22-11460] to the PLOS ONE journal. We appreciate you and the reviewers for giving us your precious time in reviewing our paper and providing valuable and constructive comments. We have been carefully incorporated the comments and suggestions provided by the reviewers and academic editor, and tried our bests to address all of them. We hope the manuscript after careful revisions meet PLOS ONE high standards and requirements.

Below is point-by-point responses to reviewer comments and we have highlighted yellow to the changes within the manuscript.

#Reviewer1: 

Thank you so much for your detailed review of our manuscript, and for your constructive comments to our manuscript. We have tried our bests to incorporate them as follows.

1. The information related to study design of EMDHS 2019 was written under Sampling Techniques section.

2. We have included information about the software we have used for analysis

3. The table was restricted to Odd Ratio, its confidence interval and p-values only

4. We have removed the empty brackets.

#Reviewer2:

Thank you so much for your detailed review of our manuscript, and for your suggestions about the usefulness of the manuscript for policymakers. By saying this, we have tried our bests to check the spelling and grammar.

---

## [Editor Report · Decision Letter 1]

14 Dec 2022

Determining factors of Neonatal Mortality in Ethiopia: An investigation from the 2019 Ethiopia Mini Demographic and Health Survey

PONE-D-22-11460R1

Dear Dr. Abay Sahile,

We’re pleased to inform you that your manuscript has been judged scientifically suitable for publication and will be formally accepted for publication once it meets all outstanding technical requirements.

Kind regards,

Sidrah Nausheen, FCPS

Academic Editor

PLOS ONE
---

## [Editor Report · Acceptance letter]

19 Dec 2022

PONE-D-22-11460R1 

Determining factors of Neonatal Mortality in Ethiopia: An investigation from the 2019 Ethiopia Mini Demographic and Health Survey 

Dear Dr. Sahile:

I'm pleased to inform you that your manuscript has been deemed suitable for publication in PLOS ONE. Congratulations! Your manuscript is now with our production department. 

Kind regards, 

on behalf of

Dr. Sidrah Nausheen 

Academic Editor

PLOS ONE